# Eccentric Overload Flywheel Training in Older Adults

**DOI:** 10.3390/jfmk4030061

**Published:** 2019-08-22

**Authors:** Kelsi Kowalchuk, Scotty Butcher

**Affiliations:** 1Health Sciences Graduate Program, University of Saskatchewan, Saskatoon, SK S7N 2Z4, Canada; 2School of Rehabilitation Science, University of Saskatchewan, Saskatoon, SK S7N 2Z4, Canada

**Keywords:** eccentric, overload, flywheel, adaptations, seniors

## Abstract

Age-related reductions in muscle strength and muscle power can have significant adverse effects on functional performance in older adults. Exercise training has been shown to be a potent stimulus for improvements in strength and power. However, investigation into how to best optimize training-related adaptations, as well as the accessibility of training methods, is needed. Traditional (TR) methods using gravity-dependent free-weights or weight machines can improve and maintain strength and power but are limited in their ability to provide constant muscle tension and high levels of muscle activation throughout the lowering (eccentric) phase of lifting. Eccentric overload (EO) training may overcome these limitations and has been shown to result in potent adaptations in both young and older adults. Methods of producing EO are significantly limited from a practical perspective. The addition of whole-body flywheel training equipment provides a practical method of producing EO and may be appropriate for older adults wanting to optimize training outcomes. Our review provides limited evidence of the use of eccentric overload flywheel training as a novel training method in seniors. Through the review of literature, EO training overcame some of the limitations set forth by traditional resistance training and demonstrated to have key benefits when combating age-related changes affecting muscle strength and muscle power. It can be concluded that EO training is an important addition to the training arsenal for older adults. Flywheel training provides a practical method of achieving EO, increasing strength and power, combating age-related adaptations, and overall improving quality of life in older adults.

## 1. Introduction

With an aging population, it is important to investigate methods that promote healthy aging to stave off the usual regressions of various body systems that affect an individual’s ability to perform daily tasks, their functional ability, and quality of life [1,2,3]. Muscle strength and power are commonly reduced with aging [3], and these reductions have a large impact on activities of daily living and functional ability [4,5]. 

Muscle strength is the ability of a muscle group to generate force against a resistance/load [6,7]. Strength declines during the aging process and is critical for day-to-day life as well as functional ability [7]. When adequate muscle strength is present, good stability, posture, function, and mobility are usually also present. Muscle strength can be assessed both concentrically and eccentrically [7,8]. Concentric contractions involve control of muscle shortening, while eccentric contractions involve controlling the contraction of a lengthening muscle. Previous research suggests that eccentric muscle strength declines with age to a lesser degree than other types of muscle strength and may, therefore, play a large role in the prevention of disability [7,9]. 

Muscle power is similar to muscle strength in its role on functional performance, but its measurement may be even more crucial in indicating declining levels of function [7,10]. Muscle power is the ability to generate force rapidly and, therefore, includes both force and speed [7,10]. Muscle power predicts functional performance as it is associated with daily movements such as climbing stairs, rising from a chair, and walking [11,12,13,14]. Similar to muscle strength, eccentric muscle power decreases at a lesser rate in comparison to concentric muscle power, highlighting its importance in an aging population [7]. 

Traditional strength and power training are common among the young, athletic population and is a method utilized when building and maintaining muscle strength and power. These training methods may also stimulate muscle hypertrophy which increases muscle mass. An increase in muscle mass increases stability and functional performance [7]. Benefits regarding strength and power training for an older population have also been documented in the literature. Strength training has been found to improve cardiovascular health, to combat sarcopenia, to reverse the loss of muscle mass and muscle strength, and overall, to combat frailty and falling incidences [15]. Although multiple benefits of traditional (free weight or machine) strength and power training that assist in maintaining functional ability are well documented, this type of resistance training is limited to selecting loads based upon the concentric action of a contraction [5,6,7]. 

Maximal eccentric strength and power are physiologically higher than those that can be produced concentrically and have been shown to be somewhat preserved as with aging [7]. Therefore, training which stimulates a higher load than can be produced concentrically (i.e. eccentric overload (EO) training) may represent a potent stimulus for adaptation in older adults. Therefore, the purpose of this article is to examine age-related adaptations, their effect on muscle strength and power, and the exploration of EO training and the benefits it may pose for a senior population. This article aims to review the literature regarding EO training and the various ways this modality can be conducted practically in a senior population. In addition, we will show how EO training for older adults helps to maintain muscle strength, muscle power, functional performance, and overall quality of life. 

## 2. Literature Review

### 2.1. Search Strategy

This review article is not a systematic review, however, we documented our search strategy. We searched the local University’s online search feature which has access to multiple databases including PubMed, Medline, and SportDiscus. We used the following keywords: flywheel, isoinertial, eccentric, strength, power, seniors, aging, older adults, exercise, training, and functional performance. Articles were screened for relevance by the authors and reviewed for their applicability to the topic. For inclusion in the primary review section (2.5.4 Application of Flywheel EO Training to Older Adults), citations needed a minimum matching terms: of one of isoinertial or flywheel, one of seniors, aging, or older adults, and either exercise or training. Articles that did not include healthy older adults or included a medical diagnosis were not an intervention trial, or utilized aerobic flywheel training and not resistance training were excluded from review in Section 2.5.4.

### 2.2. Physiological Adaptations to Aging: Muscle Strength and Power

In general, age-related changes in functional performance can be described by dividing concepts under three main physiological categories, hormonal changes, neural changes, and muscular changes [7]. Although divisions can be made in order to conceptualize these adaptations, they are not exclusive [7]. The specific changes are summarized for the reader in Figure 1, but a detailed description is beyond the scope of this article. As a whole, however, these age-related changes result in the reduction in functional performance through decreases in muscle strength and power [7,16,17,18,19,20,21,22]. It is challenging to target specific aspects of these adaptations to improve upon due to their interconnectivity, therefore, finding commonalities and examining the benefits at a higher level is more effective and advantageous [3]. 

Muscle strength is dependent upon multiple factors including age, gender, muscle fiber type, motor system, muscle mass, and sarcopenia [7]. Maximum muscular strength of individuals is observed during the second or third decade of one’s life and then progressively declines [7]. A significant decline of strength from one to one and a half percent every year from the ages of 50–70 years old has been identified [7,16]. Muscle strength is critical for day-to-day life, functional performance and preventing falling [4,5,7]. When there is well-established muscle strength, good stability, posture, function, and mobility can be observed [17,18]. 

Although strength is lost through aging, eccentric strength declines at a lower rate than both isometric and concentric strength [7]. As noted throughout literature, eccentric strength is preserved to a certain degree through specific mechanical and cellular mechanisms [19], making eccentric strength important to focus upon when maintaining strength and an individual’s functional ability as with aging. 

Muscle power is dependent upon a multitude of factors including the number and the diameter of myofibrils, muscle fiber type, coordination of neurological elements, age, and gender [7]. Muscle power declines at a greater rate than muscle strength in aging individual [7,20], and has been speculated to be a greater factor in the predictability of functional ability in comparison to strength [11,12,13,14]. It is apparent that training for strength and power is important with aging. In a similar fashion to eccentric strength, eccentric muscle power decreases less in comparison to concentric muscle power [7], which emphasizes the importance of focusing on training eccentric strength and power with aging. 

### 2.3. Traditional (TR) Strength and Power Training for Older Adults

Typical strength and power training are conducted by opposing gravitational force in addition to added weight through applying methods such as dumbbells, barbells, and other free weights or exercise machines and is known as traditional resistance (TR) training [2,21]. The additional weight applied increases muscle damage, stimulating muscle hypertrophy resulting in the building of muscles, increasing muscle mass [21]. An increase in muscle mass increases stability and functional performance [18]. Benefits regarding strength and power training for an older population have been documented in the literature [10,11,12,13,22]. Strength training is essential when improving cardiovascular health, to combat sarcopenia, to reverse the loss of muscle mass and muscle strength, and overall, to combat frailty and falling incidences [15]. Other studies have produced similar beneficial results of strength training [18,20,22], with noted additional benefits such as the ability to reduce the risk of osteoporosis, the signs of symptoms of chronic disease, improve sleep, and reduce signs of depression [23]. Although multiple benefits of strength and power training which assist in maintaining functional ability are well documented, TR training is limited regarding the load based upon the concentric action of a contraction. The maximum load and the force production during the concentric action of a contraction limits the load used during exercise [19]. Greater force production of skeletal muscle during the eccentric phase of contractions in comparison to concentric have been documented [24,25]. Therefore, the load determined concentrically results in sub-maximal activation and stimulation in the eccentric action and consequently limits the potential for muscle hypertrophy, increases in muscle strength and power and overall, the benefits of improving functional ability [26]. With this knowledge, it is important to investigate eccentric-focused training modalities in an older population to show if the limitations of traditional resistance training can be overcome and if eccentric-focused training can further assist in the healthy aging process. Additionally, as highlighted earlier, it is also important to examine eccentric overload training and its possible benefits to muscle strength and power due to its presence with aging [7,19]. 

### 2.4. Eccentric Overload (EO) Training for Older Adults 

Eccentric actions are characterized as the lengthening of muscles, such that there is a stretch in the musculotendinous complex, most commonly described as a breaking force [26,27,28]. Commonly described as the motion when walking down a hill, eccentric actions are muscle lengthening under tension [28]. Eccentric overload (EO) training is training that provides a load during the eccentric phase of contraction that is greater than that which is achieved during TR, concentric-limited loading. The most common method of EO training can be achieved by providing a load greater than that which can be trained concentrically but assisting the concentric contraction. An example of this is the 2 up, 1 down method where individuals raise the weight with two limbs and lower it with a single limb. This type of training is common in practical settings but is limited to machine exercises or isolated muscle training. As will be discussed below, the development of a flywheel device allows and promotes eccentric overload (EO) training due to its ability to apply constant and unlimited resistance through all phases of contraction, which results in higher eccentric power output than with TR training [29]. As described by recent studies, the ability to apply maximum resistance through the whole range of motion shows similar or greater strength gains in comparison to traditional resistance training and may result in similar or greater muscular adaptations [30,31]. 

The practice of EO training results in several significant benefits in an elderly population. EO combats sarcopenia more so than traditional resistance training [32]. Additionally, the implementation of eccentric training improves balance, muscle force, and muscle strength [32,33] in older adults, suggesting the potential superiority of eccentric training in comparison to concentric training [34]. Further studies are required to investigate the optimal method to introduce EO training to a senior population as well as its protocols in order to confirm and optimize the benefits [27,32]. 

As described in the literature, EO training encompasses multiple key benefits for a senior population. Eccentric actions can produce greater skeletal muscle force with minimal energy expenditure [7,26,27]. It is theorized the greater force is likely due to a combination of multiple events. These physiological events include specific actions involved in cross-bridge cycling, as well as central activation strategies specific to eccentric actions [19]. Both events seem to result in better muscular output [27]. This physiological advantage is attractive for an aging population as one could exert maximal force while exerting less energy, which is also beneficial for clinical settings [28,35]. The lower metabolic demand required for eccentric actions demonstrates metabolic efficiency, leading to speculation that concentric and eccentric actions affect energy metabolism differently. It is believed that eccentric actions initiate a non-adenosine triphosphate (ATP) rupture of the actin-myosin cross-bridges, meaning less energy is required throughout the cross-bridge process. 

Evidence also indicates there is better synchronization of the motor unit through low recruitment of motor units and there is a unique neural strategy of motor control for eccentric actions [27,36]. The low oxygen demand may also be due to the high tension produced during eccentric action resulting in ischemia to the muscle [28]. In addition, there is evidence that eccentric exercise has a lower demand on the cardiovascular system [28]. In one study, it was found that both oxygen uptake and heart rate (HR) are significantly lower during eccentric actions than concentric, which held true for all age groups [37]. Another benefit displayed by EO training is the potential for increased adaptations due to repetitive eccentric actions [38]. Literature has emphasized important alterations of the cytoskeleton and microlesions of muscle fibers occur following eccentric training [27]. EO training causes more damage to initial and secondary muscle fibers, especially in older individuals [27]. Increased damage may promote muscle repair through increased gene expression and hypertrophy, improving maximal strength [27,38,39,40]. This is key when assisting in functional performance [7,26]. Eccentric strength is preserved due to age-related adaptations, which is consistent across muscle groups [9]. Eccentric strength does not decline to a similar extent of isometric and concentric strength with aging [19]. The preservation of eccentric strength is likely both biochemical and cellular as well as an accumulation of non-contractile material relating to muscle stiffness [19]. This functional reserve of eccentric strength is important when implementing training regimes to increase functional ability and applying in clinical settings. Additional studies go above muscular adaptations and describe the cognitive improvements eccentric exercise induces [41]. 

A summary of adaptations and benefits listed regarding EO training are presented in Figure 2. This suggests a potential superior ability over traditional resistance training regarding increasing muscle strength and power with aging, and over time, improving functional ability [31,33,34,35,42,43]. Both muscle strength and power are highly associated with performing daily activities and encompass all physiological declines as previously described [7]. Through understanding the speculated benefits of eccentric exercise and comprehending how this is related to functional performance, the examination of eccentric exercise methodologies is required. As well, despite the significant benefits achieved by eccentric training in aging, the challenge is to provide readily available, practical methods of EO training that do not require significant time, equipment, and cost resources. The following section will review the use of flywheel technology in exercise training and demonstrate its application to older adults.

### 2.5. Methodology of Flywheel EO Training

#### 2.5.1. Principles of Flywheel Exercise

The use of flywheel devices in resistance training has a long history [32], but it is only in the last few decades or so that these devices have been applied to various athletic, occupational, space flight, and clinical situations [32]. Contrary to TR training where the resistance is provided by gravity-dependent free weights, machines, pneumatic devices, or elastic recoil, flywheel resistance is provided by iso-inertial momentum [44]. A detailed description of the physical properties of flywheel exercise is beyond the scope of this article and can be read elsewhere [29,32,45]. However, a brief overview follows. In short, the trainee provides the muscular force concentrically through a connection to a rope, cable, or strap that is connected to and wrapped around the axis of the flywheel, thereby spinning the flywheel against its inertia throughout the entire concentric phase of contraction. Upon reaching the maximum amount of displacement of the limb(s) for the given exercise, the momentum of the flywheel provides resistance during the eccentric phase of the exercise as the strap wraps back around the axis. Due to the laws of conservation of energy, the amount of eccentric energy that is recovered is slightly less than that expended during the concentric movement, with friction and heat loss reducing the eccentric energy slightly. Thus, the overall power achieved concentrically during the movement with the flywheel is almost identical to that achieved eccentrically, but with TR training, the eccentric power is much lower [29]. The resistance and peak powers, however, can be significantly greater during the eccentric portion of the exercise, depending on when in the eccentric phase the trainee exerts the most amount of effort. Therefore, if the trainee provides most of the effort in decelerating the flywheel in a shortened range of motion eccentrically, the peak power and resistance will be much higher than that achieved concentrically. To the knowledge of the authors, however, this style of EO training has not been studied with flywheel training and requires further research. 

As mentioned previously, TR training is limited in its ability to fully engage and activate the muscles involved in an exercise throughout each repetition compared with flywheel exercise [46]. While this is true during concentric contractions, it is even more pronounced in the eccentric phase of an exercise [47]. Therefore, the constant engagement of the muscle groups involved during flywheel exercise and the practical potential for achieving EO in training have the potential for optimizing adaptations.

#### 2.5.2. Commercial Flywheel Devices and Exercise Choice 

Early research with the flywheel as a resistance training method was mostly limited to either leg pressing or knee extension exercises utilized in laboratory settings [32]. The most commonly studied laboratory device was the Yo-Yo Flywheel designed by renowned physiologist Per Tesch [31,32]. The increasing frequency of use of this training method in research has led to the development of commercially available flywheel devices which are being used in gyms, rehabilitation clinics, and even in the home [32]. Of these commercial devices, the most commonly studied are the YoYo^TM^ (nHANCE, Stockholm, Sweden), the kBox (Exxentric AB, Broma, Sweden), the VersaPulley (VersaClimber, Santa Ana, CA, USA), and the D.EVO and D.SPORT (Desmotec, Biella, Italy). 

One of the most exciting benefits of this new technology is the wide range of exercise applications. Most of these commercial products allow for multiple different exercises to be performed. For example, the squat exercise, which was not possible with early flywheel setups, is now being studied exercise with these commercial devices [48]. In addition, many of the commercial companies allow for the measurement of a certain power, resistance, range of motion, and force variables during exercise and can allow real-time feedback to the trainee and coach, trainer, or therapist with reasonable validity and reliability [44,49,50].

#### 2.5.3. Flywheel Training Adaptations 

A thorough review of the training adaptations with flywheel training is beyond the scope of this article and the reader can be referred to Suchamel et al. [45] and Tesch et al. [32] for more detail. In short, however, there are four meta-analyses to date on the topic, three of which demonstrated the superiority of flywheel training compared with TR training on muscle strength and/or hypertrophy [51,52,53], whereas one noted similarities in the training adaptations, but not superiority [54]. This issue continues to be debated [45], but it appears that flywheel training results in at least similar training adaptations compared with TR training.

#### 2.5.4. Application of Flywheel EO Training to Older Adults

Although the concept of eccentric exercise for older adults is well known, the flywheel presents an intriguing option that has a wide range of practical applications for EO [32]. Unfortunately, there only three known studies to date that demonstrate the use of flywheel technology in apparently healthy older adults. In one study, Onambele et al [31] examined 24 older adults aged 70 ± 1.3 years randomized to a flywheel knee extension training group or a TR weight training control group. Training was conducted for 12-weeks with 8–12 repetitions and progressing from 1 to 4 sets per session. Peak knee extension power increased only in the flywheel group and, despite not directly training the gastrocnemius muscle, its tendon stiffness increased by 136%. In addition, the flywheel group increased whole-body balance, which was related to the increase in gastrocnemius tendon stiffness changes [31]. 

A more recent study utilized a flywheel device to investigate the effects of flywheel training compared with a non-training control group on mobility, postural stability, power, and balance in 18 adults aged 64 ± 3.6 years [55]. Participants trained using a flywheel device for a six-week period two to three times per week with four sets of nine repetitions of the squat exercise. The flywheel training group improved mobility, muscle power, and balance compared with the non-training group [40]. In addition, this study showed that the increases in power output were related to the increases in postural stability and may, therefore, be useful in falls prevention [55]. In addition, this is the only study in older adults to date that has used the squat exercise. The ability to use the squat exercise, which mimics the sit to stand maneuver, is crucial for exercise programming for older adults since the sit to stand maneuver is associated with muscle power and the ability to perform activities of daily living [56].

The third known study using flywheel training in healthy older adults aged 68 ± 4 years (Bruseghini et al. [33]) compared an aerobic interval training protocol to flywheel leg press training. Twelve participants completed an 8-week interval protocol on a cycle ergometer followed by a four month washout period and then eight weeks of flywheel leg press training three times per week for four sets of seven repetitions per session. While interval training improved participants aerobic fitness, blood pressure, and body composition, and flywheel training did not, only flywheel training resulted in increases in quadriceps muscle strength, decreases in LDL and total cholesterol, and increased quality of life [33].

While the above studies comprise the only known evidence of flywheel EO training in healthy older adults, a few studies on older patients with stroke [41,57], and patients with Alzheimer’s Disease [58] have been conducted. The two studies on stroke patients found that flywheel leg press training increased muscle hypertrophy, strength, and power on the affected leg, as well as increasing balance, gait parameters, and functional performance [41,57]. In a group of 12 women with Alzheimer’s Disease, 12 weeks of flywheel leg press training resulted in improvements in gait performance with increases in ankle muscle contraction time [58]. 

While the studies in older adults are few, flywheel training appears to result in several important muscle, performance, and clinical adaptations. It is clear, however, that there is much work needed to refine the protocols for training, incorporate whole-body training, and increase the body of evidence in this method of training. This area represents a huge potential for future research, particularly that comparing flywheel EO training to TR training methods.

## 3. Conclusions

This review summarized age-related adaptations and how these factors interplay and overall affect muscle strength and muscle power. As discussed, both muscle strength and muscle power are factors that are key in determining functional ability, independence, and quality of life. Traditional resistance training, which is the common method used to maintain or improve muscle strength and power, is limited through the load lifted during the concentric phase of a contraction. EO flywheel training not only overcomes this limitation through the ability to apply varying and constant resistance through all phases of a contraction, but it also poses many other benefits applicable to an aging population. Unfortunately, to date, there is little research specifically examining the types of EO flywheel training in older adults, but this review highlights the potential for this modality to be used for prevention of age-related consequences on skeletal muscle and functional performance. Further investigation is required regarding the potential of EO training and the use of flywheel devices for a senior population. 

## Figures and Tables

**Figure 1 jfmk-04-00061-f001:**
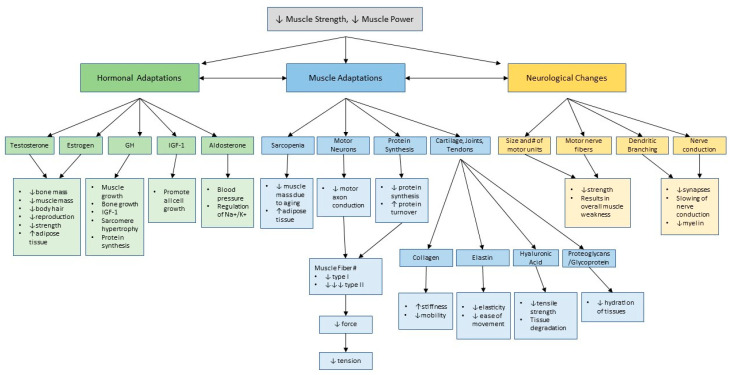
Adaptation of Aging Summary. As displayed, adaptations have been separated into components of hormonal adaptations, muscle adaptations, and neurological adaptations. Hormonal adaptations include changes in testosterone, estrogen, growth hormone (GH), insulin-growth factor 1 (IGF-1), and aldosterone. Muscle adaptations include sarcopenia, changes in motor neurons, changes in protein synthesis capabilities, and changes within cartilage, joints, and tendons. Neurological adaptations include changes in the size and number of motor units, motor nerve fibers, dendritic branching, and nerve conduction. All factors listed under their designated subgroup occur with aging and are interconnected. Information compiled from: [2,3,7,16,17,18,19,20,22].

**Figure 2 jfmk-04-00061-f002:**
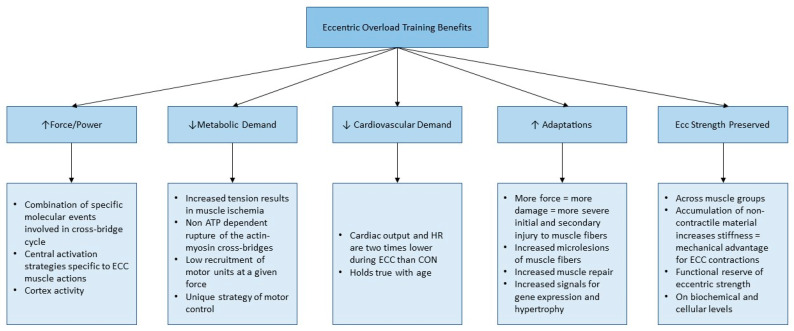
Summary of Eccentric Exercise Benefits on a senior population. The five key benefits of eccentric training on a senior population include increase force and power production, decreased metabolic demand (metabolic efficiency), decreased cardiovascular demand, increased muscular adaptations as well as the fact that eccentric strength is preserved. Information compiled from: [7,26,32,33,34,41,42,43,45,46].

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
