# Peer review of "Eccentric Overload Flywheel Training in Older Adults"

_jfmk, 2019, doi:10.3390/jfmk4030061_

Round 1
Reviewer 1 Report
Abstract
1. Can the language be more aligned with scientific principles? E.g. line 19: “Last, eccentric training proved to be a new training method investigated for a senior population as limited literature was found as well as controversial.” The word ‘proved’ is controversial in it’s uses in science. Perhaps, something along the lines of: this review provides evidence of a novel training…
2. Continuing from point 1, you seem to suggest the evidence discussed is provides proof, yet within the same sentence, you suggest the evidence is limited and controversial?
Introduction
3. Depending on your discretion, it may or may not be useful to define concentric and eccentric contractions (line 37) for the benefit of the reader. Though it may not be necessary if you expect highly informed readers. However, given that the research topic is concerning eccentric contractions, this would be apt in my opinion.
4. In line 29, for your discretionary judgement, I would suggest that you widen the statement beyond just the Canadian population, as it is widely applicable to a large number of developed nations. This may help increase the impact of this statement.
5. Starting in line 42, you state that muscle power predicts functional performance. However, strictly speaking, it does not predict. Rather, it is a reliable indicator in my opinion.
6. Line 45, typo. As should be at.
7. Line 55 needs to be referenced.
8. Although this is not a systematic review, I would suggest that the aims of article stated from line 60 are stated in a less biased tone. A review should not aim to provide evidence of something. Rather, it should aim to critically discuss the literature on the topic, and then provide discussion/conclusions following this.
Literature review
9. Line 67, you have stated three main categories to age related changes to physiology, a reference should be provided. Additionally, it would be useful to limit the statement to functional performance related physiology. Otherwise, this statement may be vulnerable in it’s broadness.
10. Line 71, line 73, line 75 need to be referenced.
11. Line 79, line 81 need to be referenced.
12. Line 86 needs to be referenced.
13. Line 94 typo, increase should be increased.
14. Line 94, please define what you mean by tension? The tension of what specifically.
15. Line 98, more specificity is needed on what type of tissue/s is/are being discussed here, muscle, all tendon, or other?
16. Line 118 needs to be referenced
17. General suggestion, would it be possible to change the word one to human throughout the paper? I feel it may read better, although this is strictly just opinion.
18. Line 142, 144 need to be referenced.
19. Line 155 needs to be referenced.
20. Line 190 needs to be referenced.
21. Line 201, reference 43 does not suggest that HR are 2 x lower. I would suggest this statement is reviewed. Additionally, as far as I am aware, the study reported oxygen uptake as opposed to cardiac output.
22. Line 205, reference 44 should be placed in the sentence before.
23. Line 208 needs to be referenced.
24. Line 211 and 212 appear to have some formatting issues regarding reference placement or presentation in text.
25. Line 240 needs to be referenced
26. Sentence starting on line 241 needs to be reviewed for grammar.
27. Line 274, Statement to the benefits to an aging population are not yet justified within this section. Alternatively, if citing previous research, please provide a reference.
28. Line 283 needs to be referenced.
29. Line 301, would you have any insight in to why the kBox being limited in exercise modality would necessitate other modalities being explored? The statement may need further justification.
30. Line 308, you have listed the difference modes on the Desmotec. However, I am not sure it is required. Further down the paper, the modes have not been discussed, compared etc.
31. Figure 2, there seems to be a duplication of the mis-statement mentioned previously regarding decreased cardiovascular demand. Additionally, I am unsure how this would be an advantage/disadvantage regarding stimulating a training response. If it is a safety concern, the specific populations for which this would be a concern need to be differentiated, i.e, previously sedentary, cardio impaired etc.
Other comments
32. It would be useful to state the magnitude of improvement following the various forms of eccentric training to allow for effective understanding of the magnitude of its effects. Additionally, it would help to compare against comparable studies that utilised traditional forms of strength and power training.
33. When citing studies pertaining to flywheel training adaptations, it would be useful to state the average ages of the groups studied.
34. Additionally, I also think that concentric loading has its unique benefits. For instance, the increased metabolic stress can provide a useful stimulus for capillarisation and mitochondrial biogenesis etc. Therefore, a more balanced discussion is required in some parts. There is also at least one meta-analysis on the topic by Vicens-Bordas et al., 2017 (https://doi.org/10.1016/j.jsams.2017.10.006) that suggests traditional gravity dependent training is not inferior to flywheel training. I feel this research needs to be critically discussed in your paper. At present, it is not mentioned within your paper.
35. Thank you for the opportunity to review this article which I enjoyed reading. I feel that with some relatively minor referencing, linguistic, and critical discussion adjustments the article is a very useful addition to the scientific literature.
Reviewer 2 Report
Manuscript title: Eccentric Overload Flywheel Training in Older Adults
Main concerns
- The authors should have diagrams to explain the three modalities of the flywheel training. The authors tried to explain how the loading and the resistance of each equipment is impose and how the eccentric exercise movement is performed or executed – this reviewer found that the explanation is inadequate or not very clear, especially if the reader is novel to the equipment. For example, what is the kMeter of the kBox? Therefore, diagrams to help illustrate that goes together with the worded explanation would be invaluable and useful, and this reviewer believe would be much, much clearer.
- This reviewer would to kindly suggest to detail out the eccentric studies conducted in the elderly cited by the authors in a table format – in this table, the authors should include the frequent, intensity (such as sets, reps, load and rest periods), time and type of the training programme that the studies have imposed onto their subjects – and highlight the key findings of each study in regards to eccentric training-induced adaptations that were observed. And of course the type of eccentric flywheel equipment used in the study.
Minor concerns
- Please do not start the sentence with a number reference like in Lines 146, 174,211, 212, 246 and many others throughout the manuscript. Write out the authors name and et al. then followed by the number references. And also, not to simply use “by” and followed by reference, for example Lines 228, 249,
- What is written from Line 34 to line 60 was almost the same as in Line 110 to 132. So please delete those sentences that are redundant and were already mentioned earlier on.
- Line 96. Elastin is not a common word and author should provide a explanation or definition of what it is.
- Line 98. “Both deceases ……….of tissues [7]”. This sentence seems awkward. Please rephase.
- Line 250. “Most studies……..benefits”. This sentence seems awkward. Please rephase.
- Line 146. “[15] note that strength training is essential when improving cardiovascular health” - does strength training can improve cardiovascular fitness. This sentence is not true in all cases.
- Line 177. “Further studies suggest…” but only one study was cited rather than several studies.
- Line 190. “better” does seem to the appropriate word here. Was thinking of “greater or higher or even optimal”.
- Line 195. “non-adenosine triphosphate (ATP) rupture of the actin-myosin cross-bridges,…” Need to explain this sentence to reader.
- Line 197. “low”? Is “less” a better word?
- Line 198. “unique neural strategy”. Please explain – very vague/unclear and seems to be oversimplified.
- Line 205. “earlier onset….” It would be nice if the actual duration is included here.
- Line 207. “initial” and ‘secondary muscle fibres”. Need to again explain what are these words to the reader.
- Line 241. The term “seniors” has not yet been defined or clarify in the study – what age range and it should be defined.
- Figure 1. The authors highlighted a couple of times throughout the manuscript that these variables are closely interlinked to each other. As such there should be arrowed lines joining the boxes horizontally in this figure.
Reviewer 3 Report
The manuscript contains several concerns that preclude it from a possible acceptance as in the present form. These are listed below.
Overall
· The Authors used “eccentric training”, “eccentric exercise”, “eccentric overload training” or “eccentric flywheel training” interchangeably. This is inaccurate. Firstly, “training” usually refers to long-term while “exercise” to acute intervention. Secondly, “eccentric training” is not equivalent to “eccentric flywheel or overload training”: the former refers usually to purely-eccentric while the latter to enhanced eccentric exercises/training. That said, please use these terms accordingly.
· The paragraph 2.1 is too generic. The Authors dealt with a number of aging-induced adaptations that would require an encyclopaedic description, and I believe this is beyond the purposes of the present manuscript. Additionally, for each topic, I would expect how the strength first and then the eccentric overload training would counteract the aging-induced effects, and according to what mechanisms. I suggest focusing on lesser topics and addressing the concern. This would mean change drastically the structure of the manuscript.
· The paragraph 2.2 is not useful here and should be incorporated within the introduction.
· Section 2.5: I don’t understand why this part is divided in sub-sections. Indeed, the flywheel technology uses the same principles and has been developed by several companies that the Authors considered separately. I would suggest rewrite this part, focusing on: 1) the different amount of inertia used in the literature in this specific population and 2) the different exercises the subjects underwent and consequently 3) the body-segments induced-adaptations.
Abstract
· Background: this must be revisited. Why eccentric overload is supposed to be beneficial? What parameters did the Authors focus on? What limitations did they mean?
· Literature review: what main including/excluding criteria were used?
· Conclusions: the conclusions are not supported by the previous paragraphs.
Introduction
· Line 29: aging is supposed to affect the worldwide and not only the Canadian population! Unless, it will be specifically stated that the literature review were designed to involve only studies including Canadian people.
· I would first introduce how strength and power affect physical capacities, and then focus on the decline in elderly population.
· Lines 38-39: 1) I believe the references provided do not support the statement. 2) Please provide a more solid rationale for this statement: is this possibly related to differences in training intervention?
· Line 47: what is meant by “traditional”?
· Lines 47-56: the Authors must clearly define what dependent parameters are they focusing on.
· The rationale of the present review is not clear at all. Both the independent and dependent parameters must be clearly stated, as well as a first mentioning of the possible inclusion/exclusion criteria.
Methods
· A clear sub-section to define the methodology of the literature review is needed.
· Lines 71-77: these sentences must be referenced.
Round 2
Reviewer 2 Report
No additional changes required.
Author Response
Thank you for your comments in making this manuscript stronger.
Reviewer 3 Report
The Authors addressed all my concerns.
Author Response

(The authors gave the same response as above.)
